# Sorafenib-Induced Apoptosis in Hepatocellular Carcinoma Is Reversed by SIRT1

**DOI:** 10.3390/ijms20164048

**Published:** 2019-08-19

**Authors:** Antje Garten, Theresa Grohmann, Katarina Kluckova, Gareth G. Lavery, Wieland Kiess, Melanie Penke

**Affiliations:** 1Center for Pediatric Research Leipzig (CPL), University Hospital for Children & Adolescents, Leipzig University, Liebigstr. 19, 04103 Leipzig, Germany; 2Institute of Metabolism and Systems Research, College of Medical and Dental Sciences, University of Birmingham, Birmingham B15 2TT, UK

**Keywords:** mitochondria, AMP-activated protein kinase, mTOR, NAMPT, sorafenib, SIRT1

## Abstract

Sorafenib is a multi-kinase inhibitor and one of the few systemic treatment options for patients with advanced hepatocellular carcinomas (HCCs). Resistance to sorafenib develops frequently and could be mediated by the nicotinamide adenine dinucleotide (NAD)-dependent deacetylase sirtuin (SIRT)1. We aimed to test whether sorafenib efficacy is influenced by cellular NAD levels and NAD-dependent SIRT1 function. We analyzed sorafenib effects on apoptosis induction, NAD salvage, mitochondrial function, and related signaling pathways in HCC cell lines (HepG2, Hep3B, und HUH7) overexpressing SIRT1 or supplemented with the NAD metabolite nicotinamide mononucleotide (NMN) compared to controls. Treatment of HCC cell lines with sorafenib dose-dependently induced apoptosis and a significant decrease in cellular NAD concentrations. The SIRT1 protein was downregulated in HUH7 cells but not in Hep3B cells. After sorafenib treatment, mitochondrial respiration in permeabilized cells was lower, citrate synthase activity was attenuated, and cellular adenosine triphosphate (ATP) levels were decreased. Concomitant to increased phosphorylation of adenosine monophosphate (AMP)-activated protein kinase (AMPK), sorafenib treatment led to decreased activity of the mechanistic target of rapamycin (mTOR), indicative of energy deprivation. Transient overexpression of SIRT1, as well as NAD repletion by NMN, decreased sorafenib-induced apoptosis. We can, therefore, conclude that sorafenib influences the NAD/SIRT1/AMPK axis. Overexpression of SIRT1 could be an underlying mechanism of resistance to sorafenib treatment in HCC.

## 1. Introduction

The multikinase inhibitor sorafenib is one of the few systemic treatment options for advanced hepatocellular carcinoma (HCC) and exerts its anti-proliferative and angiogenesis action by blockage of the Rapidly Accelerated Fibrosarcoma (RAF) and vascular endothelial growth factor (VEGF) signaling and promises better survival for patients with advanced hepatocellular carcinoma [1]. Sorafenib was also shown to attenuate mitochondrial function in HCC cells [2]. However, not all patients respond to sorafenib. Sensitizing HCC cells to sorafenib treatment by targeting metabolic pathways was shown to be a promising approach to develop novel therapeutic options for patients with advanced HCC [3,4]. Sirtuin 1 (SIRT1) is overexpressed in 55% of patients with HCC, which is associated with a higher tumor grade and poor long-term survival. SIRT1 can modulate the activity of proteins important for oncogenesis and therefore, support tumor proliferation and progression as well as drug resistance [5,6]. SIRT1 is a nicotinamide adenine dinucleotide (NAD)-dependent deacetylase. Due to metabolic alterations, cancer cells have a higher demand for NAD as a redox partner in cellular energy metabolic reactions. NAD-dependent enzymes, such as SIRT1, regulate critical cellular processes necessary for cancer cell growth, including transcription, cell-cycle progression, anti-apoptosis, and DNA repair. To provide enough NAD for cancer cells, many tumor types overexpress nicotinamide phosphoribosyltransferase (NAMPT), a key enzyme of mammalian NAD salvage [7,8]. In a previous study, we could show that inhibition of NAMPT activity via its specific inhibitor FK866 induces energy stress leading to apoptosis in HCC cell lines which was associated with adenosine monophosphate (AMP)-activated protein kinase (AMPK)-mediated inhibition of the mechanistic target of rapamycin (mTOR) signaling [9]. Based on our data and given the fact that SIRT1 could play a role in drug resistance related to its ability to deacetylate tumor suppressors, such as p53 [10], the current study aimed to find out whether sorafenib influences the NAMPT/SIRT1/AMPK signaling pathway and energy metabolism. We hypothesized that inhibition of SIRT1 supports apoptosis in hepatocarcinoma cell lines. We could show that sorafenib treatment of HCC cells led to mitochondrial dysfunction and decreased NAD and adenosine triphosphate (ATP) levels. This was associated with the activation of AMPK and subsequent inhibition of mTOR signaling. Whereas SIRT1 inhibition did not sensitize hepatocarcinoma cell lines to sorafenib, SIRT1 overexpression led to decreased apoptosis after sorafenib treatment via increasing deacetylation of p53 and could, therefore, be a potential mechanism for HCC cells to acquire sorafenib resistance.

## 2. Results

### 2.1. Sorafenib Decreases Phosphorylation of ERK and Induces Apoptosis in Hepatocarcinoma Cell Lines

To check whether sorafenib is active in the hepatocarcinoma cell lines used in this study, HUH7, Hep3B, and HepG2 cells were stimulated with different concentrations of sorafenib ranging from 0 to 10 µM for 24 h. Phosphorylation of extracellular signal regulated kinase (ERK), a downstream target of rapidly accelerated fibrosarcoma (RAF), cell viability, apoptosis, and lactate dehydrogenase (LDH) release were analyzed. Phosphorylation of ERK was decreased (*p* < 0.05 in Hep3B and HUH7 cells) after stimulation with 5 µM sorafenib (Figure 1A). Sorafenib (5 µM and 10 µM) reduced cell viability in both HUH7 and Hep3B (Figure 1B). With a maximum dose of 5 µM sorafenib, apoptotic cell number (*p* < 0.05, Figure 1C) and LDH release (*p* < 0.05, Figure 1D) was significantly increased in Hep3B and in HUH7 cells. The same results were seen in HepG2 cells (Appendix A), indicating that sorafenib is active and promoting cell death. 

### 2.2. Sorafenib-Induced Decrease of NAD Levels in Hepatocarcinoma Cells Is Not Mediated by NAMPT or SIRT1 

Next, we aimed to determine whether sorafenib treatment influenced the NAMPT/NAD/SIRT1 pathway. Interestingly, while NAMPT protein and enzyme activity were not altered (Figure 2A,B), intracellular NAD levels were significantly reduced in both HUH7 and Hep3B cell lines after stimulation with 5 µM sorafenib (*p* < 0.05, Figure 2C). While SIRT1 protein abundance was stable in sorafenib-treated Hep3B cells, HUH7 cells showed significantly reduced SIRT1 protein levels after incubation with sorafenib (*p* < 0.05, Figure 2D). Micro-RNA-34a is one candidate regulator of SIRT1 protein abundance [11] and was found upregulated in HUH7 cells dependent on sorafenib dose (*p* < 0.05, Figure 2E). We, therefore, chose HUH7 cells for further examination of sorafenib-SIRT1 interactions.

### 2.3. Sorafenib Leads to Energy Deprivation in HUH7 Cells 

We next asked whether sorafenib treatment had an impact on mitochondrial function and cellular ATP levels. Mitochondrial oxygen consumption in permeabilized HUH7 cells during different respiratory states was significantly reduced after incubation with 1 µM sorafenib for 24 h (Figure 3A). Citrate synthase activity in sorafenib-treated cells was reduced as well, indicating a decrease in mitochondrial mass (Figure 3B). To test whether sorafenib exerted a direct effect on mitochondria, sorafenib was added directly to mitochondrial preparations. Respiration was measured during the induction of different respiratory states [12,13]. The proportion of complex I-linked O_2_ consumption during oxidative phosphorylation with the substrates malate and glutamate (MG_P_) was reduced while an increase in complex-I linked leak respiration on substrates malate and glutamate (MG_L_, without the addition of adenosine diphosphate ADP) was seen. This indicates a direct effect of sorafenib on complex I (Figure 3C). ATP levels were decreased to 0.3 ± 0.1 fold after 24 h stimulation with 5 µM sorafenib (Figure 3D). This energy-deprived state was also reflected by AMPK activation and subsequent downregulation of mTOR signaling (Figure 3E). These results were verified in Hep3B cells. ATP levels were not detectable after 5 µM sorafenib treatment (Appendix A). Consequently, AMPK was found activated, and mTOR signaling was downregulated (Appendix A). These data suggest that sorafenib disturbs cellular energy metabolism by directly interfering with mitochondrial complex I function. 

### 2.4. Inhibition of SIRT1 Does Not Sensitize Hepatocarcinoma Cells to Sorafenib

One previous study showed that patients with higher SIRT1 expression in HCCs respond poorly to sorafenib [5]. Thus, we hypothesized that the inhibition of SIRT1 activity could sensitize HCC cells to sorafenib. Therefore, HUH7 cells were co-stimulated with FK866 (10 nM), a specific NAMPT inhibitor, which leads to decreased NAD levels [14,15] and consequently, lower SIRT1 activity. Unexpectedly, 10 nM FK866 in combination with sorafenib (5 µM) did not increase apoptotic cell number (Figure 4A). Neither did co-stimulation with FK866 and sorafenib increase apoptotic cell number in Hep3B cells (Appendix A). To verify this result, SIRT1 protein levels were downregulated by siRNA, and cell cycle distribution was measured. Sorafenib increased the apoptotic cell number (subG1-phase) in control cells, but no further increase was detected in SIRT1 knockdown cells (Figure 4B), indicating that downregulation of SIRT1 does not augment sorafenib-induced apoptosis. No other differences in cell cycle distribution between SIRT1 knockdown and control cells were found (Figure 4B). 

### 2.5. SIRT1 Overexpression and NMN Supplementation Suppress Apoptosis in HUH7 Cells

Vice versa, SIRT1 protein was overexpressed, or NAD levels were normalized by the NAMPT enzyme product nicotinamide mononucleotide (NMN). SIRT1 overexpression and supplementation with NMN both decreased sorafenib-induced apoptosis in HUH7 cells. SIRT1 overexpression had a significantly greater anti-apoptotic effect than NMN supplementation (*p* < 0.05, Figure 5A). NMN treatment led to a normalization of NAD levels after sorafenib treatment (Figure 5B) and decreased phosphorylation of AMPK (Figure 5C). mTOR and ERK signaling, however, was neither influenced by NMN treatment (Appendix A) nor by SIRT1 overexpression (Appendix A). Similarly, SIRT1 overexpression neither counteracted sorafenib-induced mitochondrial dysfunction (Appendix A) nor influenced forkhead box o (FOXO)-mediated expression of the antioxidative enzymes manganese superoxide dismutase (MnSOD) or catalase (Appendix A). 

SIRT1 overexpression, however, counteracted polymerase 1 (PARP-1) cleavage (Figure 6A), indicating an anti-apoptotic action. In line with this, deacetylation of p53 was increased in SIRT1-overexpressing cells (Figure 6A,B), and total p53 levels were reduced (Figure 6A,C). The expression of the pro-apoptotic factor p53 upregulated modulator of apoptosis (PUMA) is regulated by p53 [16] and was found to be slightly decreased in sorafenib-treated SIRT1 overexpressing cells compared with vector-transfected cells (Figure 6A,D). The p53-PUMA pathway could, therefore, be involved in the anti-apoptotic effect induced by SIRT1.

## 3. Discussion

One of the few systemic treatment options for advanced HCCs is the multi-kinase inhibitor sorafenib, which, however, does not lead to therapy success in all patients. One reason could be the aberrant expression of the NAD-dependent deacetylase SIRT1, which regulates cancer drug resistance [18]. It has been shown that SIRT1 overexpressing HCCs are more resistant to sorafenib, grow more aggressively and that patients with these tumors have a poorer prognosis compared to patients with non-SIRT1 overexpressing tumors [19]. Based on these findings, we asked whether modification of SIRT1 protein or activity could alter tumor cell response to sorafenib in HCC cell lines. 

Sorafenib targets several central processes in tumor development and progression. Specifically, sorafenib is able to inhibit tyrosine kinases of the VEGF signaling pathway to reduce tumor angiogenesis and RAF kinase, associated with an inhibition of the MAPK/ERK pathway leading to reduced cell proliferation [20]. This is in line with our study, showing that all cell lines we used responded with apoptosis to sorafenib treatment. Other studies showed that sorafenib induces the intrinsic pathway of apoptosis via mitochondrial translocation of Bax and release of cytochrome c [21,22]. This is in line with our results showing that sorafenib induces mitochondrial dysfunction and a decrease in cellular ATP levels. 

SIRT1 deacetylates proteins and regulates their transcription or activity having an impact on cellular energy metabolism as well as stress response pathways [23]. SIRT1 activity depends on NAD levels and therefore, on NAD salvage by NAMPT [21]. Interestingly, while NAMPT protein abundance and activity was not altered by sorafenib incubation, NAD levels were significantly downregulated in the HCC cell lines used in our study. We did not find evidence for increased activity of both NAD consuming enzymes poly (ADP-ribose) polymerase 1 (PARP-1), and SIRT1, which was not altered in Hep3B and decreased in HUH7 after incubation with sorafenib, potentially via p53-dependent upregulation of micro-RNA-34a [11,24]. Interestingly, we showed an increase in microRNA-34a in HUH7 cells after incubation with sorafenib. In rats with hepatocellular carcinoma, rno-miR-34a-5p was upregulated after sorafenib treatment in lung metastasis [25]. Furthermore, micro-RNA-34a was shown to sensitize hepatocarcinoma cells to sorafenib treatment [26]. Upregulation of microRNA-34a by sorafenib could, therefore, be a possible underlying mechanism of SIRT1 downregulation in HUH7 cells.

We showed that incubation with sorafenib induced a reduction in mitochondrial oxygen consumption, which has been described in cardiac myocytes and HCC cells [2,27,28,29]. In contrast to another study [30], we found a direct inhibition of complex I activity by sorafenib in addition to a decrease in mitochondrial mass as detected by lower citrate synthase activity in sorafenib-treated cells. In concordance with inducing mitochondrial dysfunction, sorafenib decreased ATP levels in hepatocarcinoma cells and led to the activation of the cellular energy sensor AMPK. AMPK is less phosphorylated and therefore, less active in hepatocarcinoma cell lines compared to primary human hepatocytes [15]). Active AMPK inhibits phosphorylation of mTOR and its downstream targets p70S6K and 4EBP1, which are important regulators of cell proliferation and protein biosynthesis, respectively [31]. This is in contrast to another study which showed that sorafenib did not affect AMPK activation or ATP levels, which can be explained by the application of a much lower dose of sorafenib (0.1 µM) [30]. Interestingly, activation of AMPK by retinoic acid sensitized HepG2 cells to sorafenib treatment [32].

Previous studies showed that high SIRT1 levels in HCC tissue of patients are associated with a worse outcome and an increased resistance to sorafenib [5,19]. We, therefore, asked whether downregulation of SIRT1 would sensitize HCC cells to sorafenib. FK866 is a specific NAMPT inhibitor which decreases NAD levels associated with lower SIRT1 activity and induces apoptosis in HCC cell lines [9]. A combination of FK866 with etoposide sensitizes leukemia cell lines to etoposide treatment by acetylation and subsequent accumulation of p53 [33]. Co-stimulation of FK866 and sorafenib, however, did not increase apoptotic cell number in our cell model. Similarly, downregulation of SIRT1 protein did not lead to increased apoptosis. This could be explained by the fact that sorafenib treatment already decreased NAD levels and SIRT1 protein abundance in our model, with additional downregulation of SIRT1 protein or activity having no effect.

Interestingly, increasing SIRT1 protein abundance by overexpression or enhancing SIRT1 activity by supplementing NMN, the enzyme product of NAMPT, counteracted sorafenib-induced apoptosis in HUH7 cells. This is in line with the human studies mentioned above. Moreover, co-treatment of NMN and sorafenib normalized NAD levels and AMPK phosphorylation, whereas mTOR signaling or ERK phosphorylation was not influenced. The rat sarcoma (RAS)/ERK and phosphatidylinositol-4,5-bisphosphate 3-kinase (PI3K)/mTOR pathways are two main signaling pathways controlling cell survival and proliferation which do not run side by side, but rather regulate each other by cross-inhibition or activation. Since mTOR is a key signal receiver from the RAS/ERK pathway, it could be possible that the RAS-ERK pathway and not AMPK controls mTOR activity in our cell system [34]. Similarly, SIRT1 overexpression did not improve mitochondrial function, which is in line with a direct effect of sorafenib on complex I. Potentially, cells adapt to complex I inhibition by increasing glycolysis as is described for SIRT1 overexpressing cells [35] and increased AMPK phosphorylation [36], which is in line with our results. Normalization of glycolytic flux would also explain the observed anti-apoptotic effect of NAD repletion via NMN.

p53 is known as a tumor suppressor protein which is mutated in many cancer types. SIRT1 inhibits p53 activity by deacetylation, accompanied by increased proliferation and anti-apoptotic effects [37,38,39]. SIRT1 regulates p53 transcription-dependent apoptosis, which requires the expression of apoptosis-related target genes, including BAX, p53 upregulator (PUMA), and NOXA [40]. We observed sorafenib-induced upregulation of PUMA, but not of BAX or BCL_XL_ [41]. SIRT1, however, can also regulate p53 transcription-independent apoptosis, which would lead to cytochrome c release from mitochondria [42]. This is mediated by increased oxidative stress, a condition induced by sorafenib [27,43,44].

## 4. Material and Methods

### 4.1. Cell Culture and Treatment

Cell lines were obtained from Leibniz Institute DSMZ-German Collection of Microorganisms and Cell Cultures, Braunschweig, Germany. HepG2 cells and Hep3B cells were cultured in MEM medium. HUH7 cells were grown in DMEM medium with high glucose. All media were supplemented with 10% fetal bovine serum (FBS) and 2 mM glutamine. All cells were maintained at 37 °C in a humidified atmosphere of 95% air and 5% CO_2_. For experiments, cells were seeded in appropriate cell culture plates and grown for 24 h. Afterward, cells were starved overnight in serum-free medium (SFM) without 10% FBS. Cells were stimulated with sorafenib (1 µM; 2.5 µM, and/or 5 µM), nicotinamide mononucleotide (NMN) (250 µM), FK866 (10 nM) and solvent control (DMSO) for 24 h or as stated below. FK866 (Sigma-Aldrich, St. Louis, MO, USA) and sorafenib (Cayman Chemicals) were dissolved in DMSO (stock solution: 10 mM). NMN (Sigma-Aldrich) was dissolved in the appropriate medium (stock solution: 100 mM).

### 4.2. Transfection

HUH7 cells were split 1:3 the day before transfection. Cells were transfected with pECE-Flag-SIRT1 (1 µg DNA/0.5 × 10 ^6^ cells) or the empty vector using the NEON Transfection System (Invitrogen/Thermo Scientific, Waltham, MA, USA). The next day, the medium was changed to SFM, and cells were starved overnight. Cells were stimulated, as stated above.

### 4.3. Cell Viability and Apoptosis

Cell viability was measured by using the Cell Proliferation Reagent WST-1 (Roche Diagnostics GmbH, Mannheim, Germany) as described previously [9]. FITC Annexin V Apoptosis Detection Kit (BD Pharmingen™, Franklin Lakes, NJ, USA) was appropriated to determine apoptotic cells as described previously [9]. As a positive control, cells were stimulated with palmitate (0.5 mM) for 24 h.

### 4.4. Lactate Dehydrogenase (LDH) and Adenosine Triphosphate (ATP) Measurement

LDH release was measured using Pierce™ LDH Cytotoxicity Assay Kit (Thermo Scientific, Waltham, MA, USA), and ATP levels were measured by CellTiter-Glo Luminescent Cell Viability Assay (Promega, Madison, WI, USA) according to the manufacturer’s protocol.

### 4.5. Cell Cycle Analysis

Adherent and floating cells were collected after 24 h. After pelleting, cells were fixed in 70% ethanol overnight, followed by incubation with RNAse (30 mg/mL) and propidium iodide (PI) and finally measured by flow cytometry (BD LSR II) [45].

### 4.6. Protein Extraction and Western Blot Analyses

Cells were lysed in modified RIPA buffer as previously described [46] after stimulation with indicated substances after 24 h. Protein concentration was determined using the Pierce BCA protein assay (Thermo Scientific, Waltham, MA, USA). Thirty micrograms proteins were separated by SDS-PAGE and subsequently blotted to nitrocellulose membranes. Membranes were blocked in 5% non-fat dry milk in TBS buffer containing 0.1% Tween 20 and incubated with the following antibodies: anti-phospho-ERK, anti-ERK, anti-phospho-AMPKα (Thr172), anti-AMPKα, anti-phospho-mTOR (Ser2448), anti-mTOR, anti-phospho-4E-BP1 (Ser65), anti-4EBP1, anti-phospho-p70S6K (Thr389), anti-S6K, anti-phospho-FOXO1 (Ser256), anti-SOD2, anti-Catalase, anti-PARP, anti-acetyl-p53 (Lys382), anti-p53, anti-PUMA, anti-SIRT1, anti-αTUBULIN (Cell Signaling, Danvers, MA, USA), anti-Visfatin (Bethyl Laboratories, Montgomery, TX, USA), and anti-GAPDH (Merck Millipore, Darmstadt, Germany) according to the manufacturer’s instructions. Detection of proteins was carried out using Luminata Classico Western HRP Substrate (Merck Millipore) or Amersham ECL Prime Western Blotting Detection Reagent (GE Healthcare, Chicago, IL, USA).

### 4.7. NAMPT Enzyme Activity

NAMPT enzyme activity was measured as described previously [15]. Briefly, cells were incubated with the indicated substances, harvested and lysed in sodium phosphate buffer, pH 7.4. After adding the reaction buffer containing TRIS, ATP, PRPP, MgCl, and radiolabeled ^14^C-nicotinamide, the mixture was transferred to acetone-pre-soaked glass microfiber filters (Whatman). ^14^C-labeled nicotinamide was converted to ^14^C-NMN. Radioactivity of ^14^C-NMN was quantified in a liquid scintillation counter in counts per minute (cpm) (Wallac 1409 DSA, PerkinElmer, Rodgau, Germany) [15].

### 4.8. Intracellular NAD Levels

NAD levels were measured by reversed-phase HPLC as described previously [9]. After 24 h of stimulation with the indicated substances, cells were lysed in 1 M perchloric acid and neutralized with 3 M K_2_CO_3_. After centrifugation, the supernatant was loaded into an HPLC system using a gradient HPLC method with UV detection [9].

### 4.9. RNA Extraction and Real-Time qPCR

Total RNA was extracted by RNeasy Mini Kit (Qiagen, Hilden, Germany). according to the manufacturer`s protocol. One microgram of total RNA was transcribed into cDNA by M-MLV Reverse Transcriptase (Invitrogen). Afterward, Taqman^®^ or SybrGreen^®^ analyses were performed using the qPCR Master Mix Plus Low ROX (Eurogentec, Seraing, Belgium) or 2X Takyon for SYBR Assay-ROX, respectively, and the Applied Biosystems 7500 Real Time PCR System. The following primer and probe pairs were used: NAMPT (forward: 5′-GCA-GAA-GCC-GAG-TTC-AAC-ATC-3′; reverse: 5′-TGC-TTG-TGT-TGG-GTG-GAT-ATT-G-3′; probe: 5′-TGG-CCA-CCG-ACT-CCT-ACA-AGG-TTA-CTC-AC-3′), β-actin (forward: 5′-CGA-GCG-CGG-CTA-CAG-CTT-3′; reverse: 5′-CCT-TAA-TGT-CAC-GCA-CGA-TTT-3′; probe: 5′-ACC-ACC-ACG-GCC-GAG-CGG-3′), TATA-box-binding protein (TBP; forward: 5′-TTG TAA ACT TGA CCT AAA GAC CAT TGC-3′: reverse: 5′-TTC GTG GCT CTC TTA TCC TCA TG-3′; probe: AAC GCC GAA TAT AAT CCC AAG CGG TTT G-3′) and hypoxanthine phosphoribosyltransferase (HPRT; forward: 5′- GGC AGT ATA ATC CAA AGA TGG TCA A-3′; reverse: 5′-GTC TGG CTT ATA TCC AAC ACT TCG T-3′; probe: 5′- CAA GCT TGC TGG TGA AAA GGA CCC C-3′).

After reverse transcription with miScript II RT, micro-RNA-34a was quantified using miScript primer assays for miR34a-1 and Hs_RNU6_2_1 for normalization with miScript SYBR Green PCR kit (Qiagen, Hilden, Germany).

### 4.10. High-Resolution Respirometry

Measurements of mitochondrial oxygen consumption were done according to [12,13]. Briefly, cells were permeabilized by digitonin (10 μg/1 × 10^6^ cells). Alternatively, mitochondria were isolated as described below. Samples were added to the Oxygraph2K (Oroboros Inc., Innsbruck, Austria) chambers in Mir05 buffer (0.5 mM EGTA, 3 mM MgCl2, 60 mM K-lactobionate, 20 mM taurine, 10 mM KH2PO4, 110 mM sucrose, 1 g/L essential fatty acid-free BSA, 20 mM Hepes, pH 7.1 at 30 °C) at 37 °C and substrates for mitochondrial complex I (malate 2 mM, glutamate 10 mM), ADP (5 mM), succinate (20 mM) as substrate for complex II of the electron transport chain, uncoupler FCCP (0.25 μM sequentially until maximal uncoupling) and inhibitors for complex I (rotenone, 0.5 μM) and complex III (Antimycin A, 2.5 μM) were added sequentially to determine O_2_ flow normalized to number of cells (for permeabilized cells) or O_2_ flux normalized to mitochondrial protein amount (for mitochondrial preparations) in different respiratory states. Mitochondrial membrane integrity was assessed by measuring O_2_ flux after adding cytochrome c (10 μM) after complex I substrates and ADP.

### 4.11. Isolation of Mitochondria

Mitochondria were isolated from approximately 70 to 80 × 10^6^ cells by differential centrifugation according to [47,48] with some modifications. Cells were trypsinized and resuspended in mitochondrial isolation buffer (D-Mannitol (M4125) 225 mM, Sucrose (S0389) 75 mM, EGTA 0.1 M, pH 7.4 (E4378) 1 mM. The cell suspension was transferred to a pre-cooled glass/Teflon potter, homogenized with 30 strokes at 1000 rpm and centrifuged in pre-cooled 2 mL tubes at 600× *g* for 10 min at 4 °C. After transfer of supernatant to a new tube, cell lysates were centrifuged at 3000× *g* for 10 min at 4 °C. This was repeated with centrifugation at 7000× *g* for 10 min at 4 °C. The mitochondrial pellet was washed with 5 mL of fresh buffer and again centrifuged at 7000× *g* for 10 min at 4 °C. After discarding of supernatant, the pellet was resuspended in the small volume of buffer (approx. 50 μL), and protein concentration was determined using Bio-Rad DC protein assay. For respiration experiments, 300 to 500 μg protein/chamber was used. Sorafenib (1 μM) or DMSO (0.1%) was added directly to each chamber, and respiration was measured as described above.

### 4.12. Citrate Synthase Assay

Citrate synthase activity was determined using a colorimetric assay by measuring the conversion of 5,5′-dithio-bis-(2-nitro-benzoic acid) (DNTB) according to [49]. Briefly, cells incubated with 1 μM Sorafenib or DMSO 0.1% for 24h were trypsinized, resuspended in citrate synthase sample buffer (Hepes 20 mM, EDTA 1 mM, TritonX100 0.1% (*v*/*v*) and lysed by freeze-thawing 4× on dry ice. After centrifugation at 15,000× *g*, 10 min, 4 °C, supernatants were used for protein determination (Bio-Rad DC protein assay). Twenty micrograms of protein were added to 200 μL reaction mix (Tris 20 mM pH 7.5, DTNB 0.1 mM, acetyl-CoA 0.3 mM) and blank absorbance at 412 nm was measured for 15 min at 30 °C. The reaction was started by adding oxaloacetate (0.5 mM), and measurement was continued for 15 min. Citrate synthase activity was determined by subtracting blank values and normalization to DMSO controls, which were set 1.

### 4.13. Statistical Analyses

Statistical analyses were performed using GraphPad^®^Prism for Windows (Version 5.04, GraphPad Software, San Diego, CA, USA). Significance was tested using either one-way analysis of variance (ANOVA) followed by Bonferroni post hoc test or unpaired Student’s *t*-test. Data are shown as means ± SEM. Statistical significance was set *p* < 0.05.

## 5. Conclusions

We conclude that sorafenib targets multiple pathways, including energy metabolism and the NAD/SIRT1/AMPK axis and that increased SIRT1 protein abundance and NAD levels could be an underlying mechanism which promotes resistance to sorafenib in patients with HCC.

## Figures and Tables

**Figure 1 ijms-20-04048-f001:**
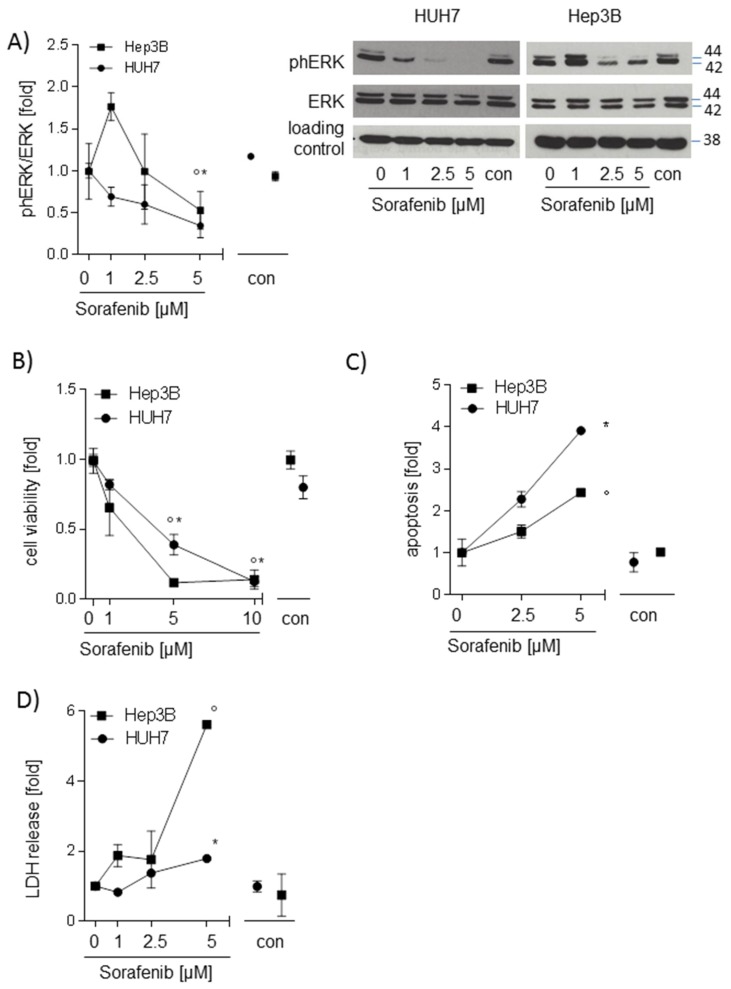
Sorafenib decreases phosphorylation of extracellular signal regulated kinase (ERK) and induces apoptosis in hepatocarcinoma cell lines (HUH7, Hep3B) (**A**) Western blot analysis of ph-ERK and total ERK after stimulation with sorafenib (1 µM; 2.5 µM, and 5 µM) for 24h. One representative blot out of three experiments is shown. Densitometric analysis was performed to semi-quantify values (*n* = 3). (**B**) The water soluble tetrazolium (WST)-1 assay was performed to measure changes in cell viability in HUH7 and Hep3B cells after treatment with sorafenib (1 µM; 5 µM, and 10 µM) (*n* = 3). (**C**) Sorafenib-induced apoptosis (2.5 µM and 5 µM) was determined by Annexin-V/Propidium iodide (PI) Apoptosis Detection Kit and flow cytometry. Annexin V-fluorescein isothiocyanate (FITC) positive cells and Annexin-V-FITC/PI double-positive cells were defined as apoptotic cells (*n* = 3). (**D**) Lactate dehydrogenase (LDH) release was measured using Pierce™ LDH Cytotoxicity Assay Kit (Thermo Fisher Scientific). The mean value of serum-free medium (0 µM) was used to normalize values of the respective datasets. Data are shown as mean ± SEM. con: solvent control (DMSO); * *p* < 0.05 (HUH7); ° *p* < 0.05 (Hep3B).

**Figure 2 ijms-20-04048-f002:**
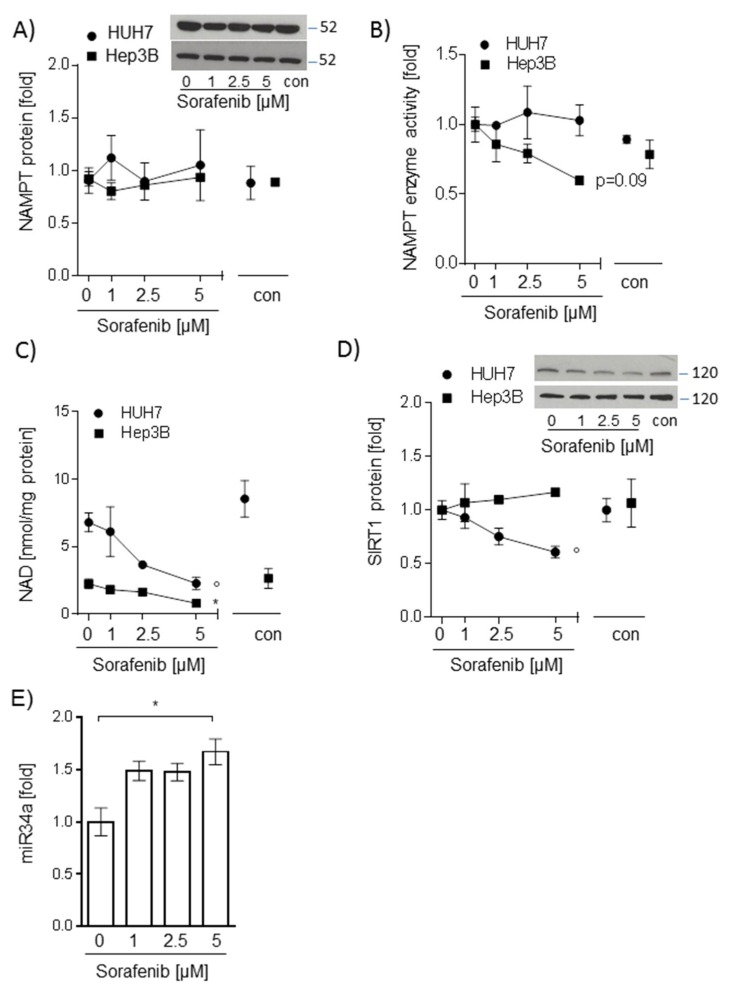
Sorafenib-induced decrease of nicotinamide adenine dinucleotide (NAD) levels in hepatocarcinoma cells is not mediated by nicotinamide phosphoribosyltransferase (NAMPT) or sirtuin (SIRT1) NAMPT (**A**), and SIRT1 (**D**) protein were analyzed by Western blot analysis. NAMPT and SIRT1 protein were normalized to glyceraldehyde-3-phosphate dehydrogenase (GAPDH) (not shown). One representative blot out of three experiments is shown as an insert. Densitometric analysis was performed to semi-quantify values (*n* = 3). (**B**) Intracellular NAMPT enzyme activity was determined by measuring the conversion of ^14^C-labeled nicotinamide to ^14^C-NMN (*n* = 3). (**C**) Intracellular NAD levels were measured by gradient HPLC/UV analysis (*n* = 3). (**E**) Expression of miRNA34a was measured using miScript primer assays for miR34a-1 and normalized to Hs_RNU6_2_1 (*n* = 3). For all experiments, HUH7 and Hep3B cells were stimulated with sorafenib (1 µM; 2.5 µM, and 5 µM) for 24 h. The mean value of serum-free medium (0 µM) was used to normalize values of the respective datasets. Data are shown as mean ± SEM. con: solvent control (DMSO); ° *p* < 0.05 (HUH7); * *p* < 0.05 (Hep3B).

**Figure 3 ijms-20-04048-f003:**
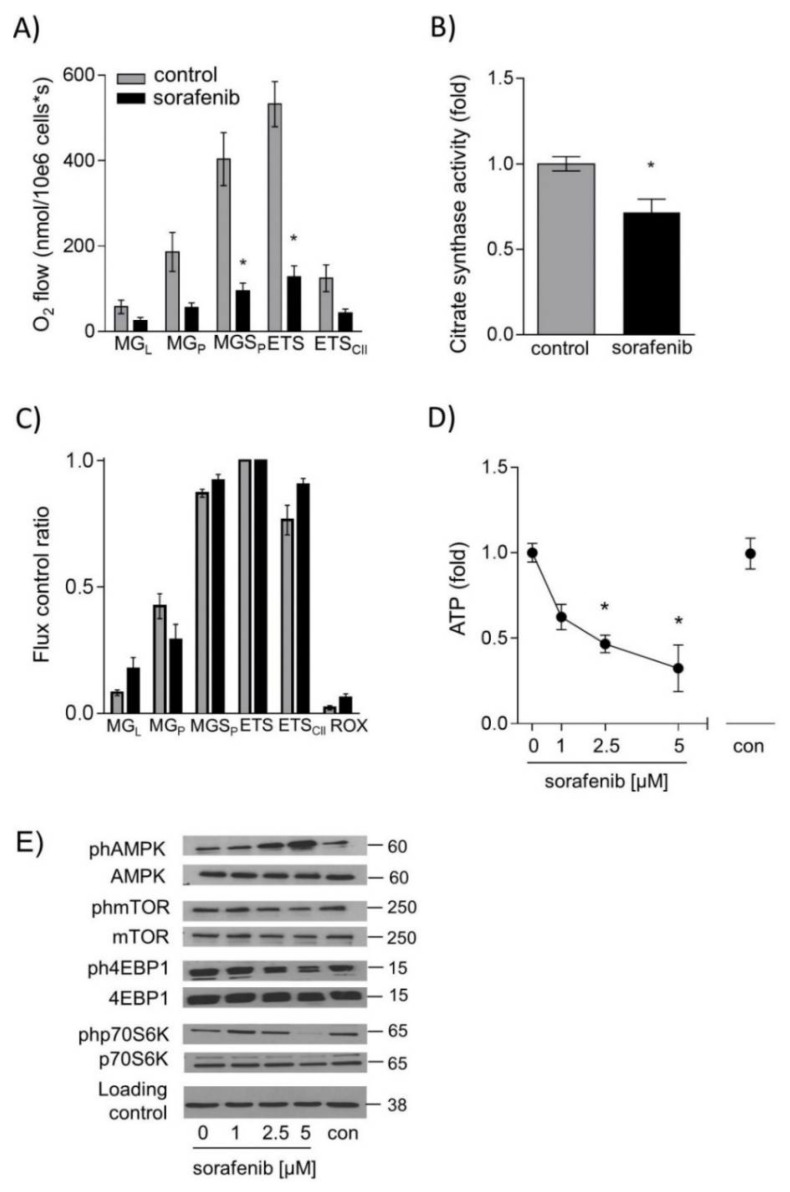
Sorafenib decreases respiration, adenosine triphosphate (ATP) levels and activation of the mechanistic target of rapamycin C1 (mTORC1) pathway in HUH7 cells. Mitochondrial respiration was measured after addition of substrates malate and glutamate (MG_L_), ADP (MG_P_), succinate (MGS_P_), uncoupling by carbonyl cyanide-4-(trifluoromethoxy)phenylhydrazone FCCP (ETS), complex I inhibitor rotenone (ETS_II_) and complex III inhibitor Antimycin A (ROX) by high resolution respirometry (Oxygraph2K, Oroboros, Austria) in (**A**) permeabilized HUH7 cells (1 µM sorafenib for 24 h) and (**C**) after direct addition of sorafenib (10 µM) to isolated mitochondria. Values are normalized to maximal O_2_ flux (uncoupling conditions). (**B**) Citrate synthase (CS) activity was determined by measuring the conversion of 5,5′-dithio-bis-(2-nitro-benzoic acid) (DNTB) 20 μg of protein were added to 200 μL reaction mix (Tris 20 mM pH 7.5, DTNB 0.1 mM, acetyl-CoA 0.3 mM) and blank absorbance at 412 nm was measured for 15 min at 30 °C. The reaction was started by adding oxaloacetate (0.5 mM), and measurement was continued for 15 min. Blank values were subtracted, and values were normalized to the mean of DMSO controls. (**D**) ATP levels were measured by CellTiter-Glo Luminescent Cell Viability Assay after stimulation with sorafenib (1 µM; 2.5 µM, and 5 µM) for 24 h. (**E**) One representative blot out of three experiments of ph-AMPK, ph-mTOR, ph-4EBP1, and ph-p70S6K is shown. Phosphorylated protein was normalized to total protein and loading control (GAPDH). The mean value of serum-free medium (0 µM) was used to normalize values of the respective datasets. Data are shown as mean ± SEM. con: solvent control (DMSO); * *p* < 0.05.

**Figure 4 ijms-20-04048-f004:**
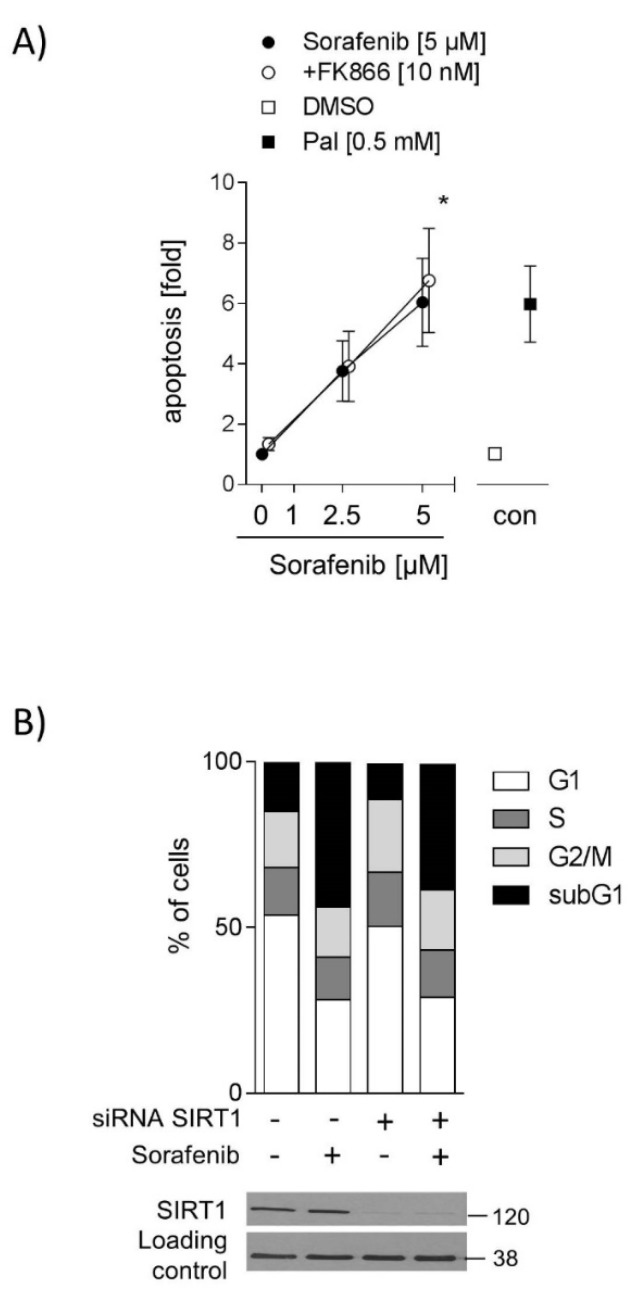
Inhibition of SIRT1 does not sensitize hepatocarcinoma cells to sorafenib (**A**) Cells were stimulated with sorafenib (2.5 µM and 5 µM) alone or in combination with FK866 (10 nM) for 24 h. Apoptosis was determined by FITC Annexin V Apoptosis Detection Kit and flow cytometry. Annexin V-FITC positive cells and Annexin-V-FITC and PI-positive cells were defined as apoptotic cells (*n* = 3). DMSO serves as solvent control and palmitate (0.5 mM) as a positive control. The mean value of serum-free medium (0 µM) was used to normalize values of the respective datasets. (**B**) SIRT1 protein levels were downregulated by SIRT1 siRNA using electroporation. Control cells were transfected with control siRNA. Cell cycle distribution was analyzed by PI staining to analyze DNA content. Data are shown as mean ± SEM. * *p* < 0.05.

**Figure 5 ijms-20-04048-f005:**
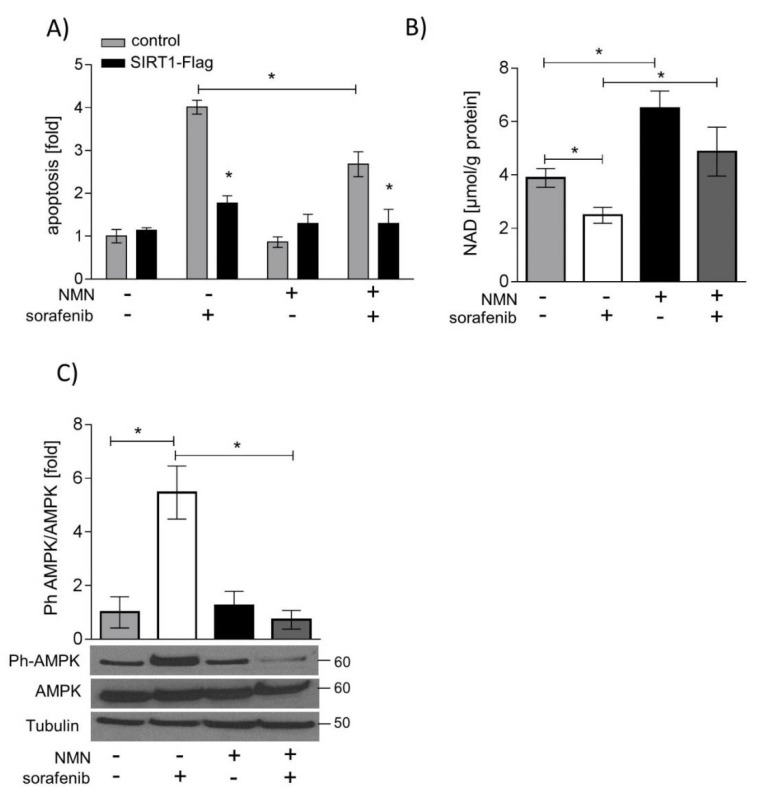
SIRT1 overexpression and nicotinamide mononucleotide (NMN) supplementation suppress apoptosis induction by sorafenib. HUH7 cells were stimulated with sorafenib (5 µM), NMN (250 µM), or a combination of sorafenib and NMN. (**A**) SIRT1 was overexpressed in HUH7 cells by electroporation. Apoptosis was determined by FITC Annexin V Apoptosis Detection Kit and flow cytometry. Annexin V-FITC positive cells and Annexin-V-FITC and PI-positive cells were defined as apoptotic cells (*n* = 3). (**B**) Intracellular NAD levels were measured by gradient HPLC/UV analysis (*n* = 3). (**C**) Phosphorylated AMPK was normalized to total AMPK and loading control (Tubulin). One representative blot out of three experiments is shown. The mean value of solvent control (DMSO) was used to normalize values of the respective datasets. One representative blot out of three experiments is shown. Data are shown as mean ± SEM. * *p* < 0.05.

**Figure 6 ijms-20-04048-f006:**
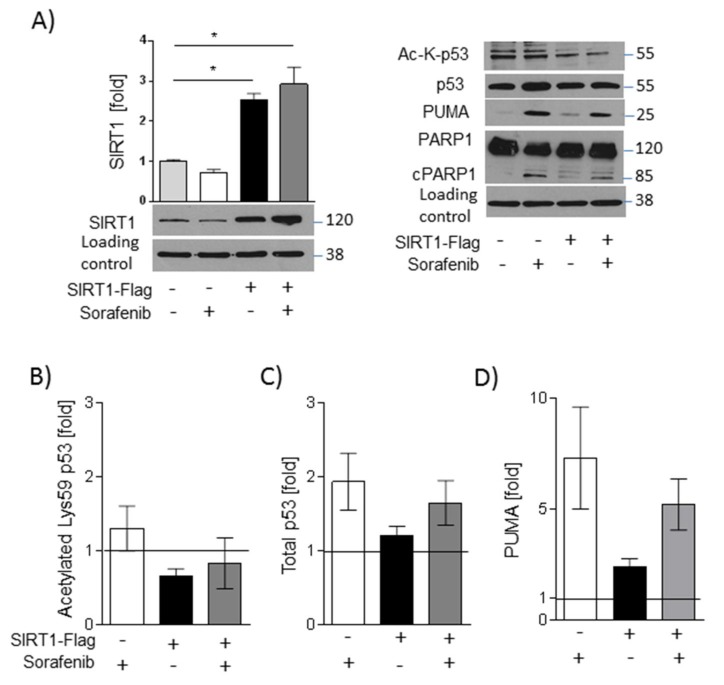
SIRT1 overexpression leads to downregulation of sorafenib-induced p53 signaling**.** HUH7 cells were stimulated with sorafenib (5 µM). SIRT1 was overexpressed in HUH7 cells by electroporation. (**A**) Left: Densitometric analysis of SIRT1 was done using ImageJ (NIH) [17]. SIRT1 was normalized to mean values of GAPDH, which served as a loading control. Right: Acetylated Lys382-p53, total p53, poly (adenosine diphosphate ribose) polymerase 1 (PARP-1), and p53 upregulated modulator of apoptosis (PUMA) was detected by Western blot analysis, GAPDH served as a loading control. One representative blot out of three experiments is shown. Densitometric analysis of (**B**) acetylated Lys382-p53, (**C**) total p53, and (**D**) PUMA was done using ImageJ (NIH). Solvent control (DMSO) was set =1, indicated by a line. One representative blot out of three experiments is shown. Data are shown as mean ± SEM. Sirt1-FLAG: overexpression plasmid for Flag-tagged SIRT1.

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
