# Peer review of "Sorafenib-Induced Apoptosis in Hepatocellular Carcinoma Is Reversed by SIRT1"

_ijms, 2019, doi:10.3390/ijms20164048_

Round 1

Reviewer 1 Report

Comment 1. The authors were asked to change the term "whether or not" into "whether" (e.g. line 66, 89, 111) (see previous comment 5). however these alterations were not executed in the revised manuscript.

Comment 2, referring to former comment 2:

The authors state in the point by point letter that they normalized  "all values of the data set by dividing through this SFM or control mean" (#2).

If the values are defined as "normalized", the SFM value should be set to "1".

Why did the authors not use a DMSO control instead of SFM control as they state the inhibitor is dissolved in DMSO e.g. in Figure 5C?

Author Response

Dear reviewers and editors,

Thank you once more for the careful evaluation of our manuscript ijms-559629. We addressed additional comments of reviewer 1 and marked changes in yellow. Please find detailed answers below. Reviewer 2 had no further comments.

Specific comments of reviewer 1:

Ø  Comment 1. The authors were asked to change the term "whether or not" into "whether"  (e.g. line 66, 89, 111) (see previous comment 5). however these alterations were not executed  in the revised manuscript.

We are really sorry for this. This was a mistake in uploading the correct version. We now changed all “whether or not” to “whether”: page 2 line 66, page 4 line 89, page 6 line 111, page 11 line 213

Ø  Comment 2, referring to former comment 2:

             The authors state in the point by point letter that they normalized "all values of the           data set   by dividing through this SFM or control mean" (#2).

             If the values are defined as "normalized", the SFM value should be set to "1".

             Why did the authors not use a DMSO control instead of SFM control as they state the     inhibitor is dissolved in DMSO e.g. in Figure 5C?

We apologise for not addressing this point clearly enough in the first rebuttal: We divided the data by the mean of either the SFM or the DMSO control of the respective experiment to adjust for variation between independent experiments, e.g. exposure of Western blots or seeding slightly different cell numbers. We chose to divide by the control means to not cancel out variation in the controls which would have happened if we set the controls to 1. Therefore, we retained standard error values which are seen in the graphs. In cases where standard errors are very small (e.g. Figure 1D) it looks like there is no standard error depicted, but it is just not visible.

In initial experiments where we tested Sorafenib dose responses (Figures 1-4) we used both SFM and DMSO controls to exclude an effect of DMSO. Since we did not detect a difference in controls we chose SFM values for normalization. In Figure 5 and 6 we used DMSO controls and normalized to the mean value of DMSO controls. We accordingly corrected the Figure legends for Figure 5 and 6.

Additionally, we saw that lines in legends for Figure 1A and Figure 2D were shifted. We corrected this and included the corrected figures.

Reviewer 2 Report

The authors have adequately addressed the reviewers' concerns.

Author Response

Thank you.

This manuscript is a resubmission of an earlier submission. The following is a list of the peer review reports and author responses from that submission.

Round 1

Reviewer 1 Report

In this manuscript, the authors investigated the effects of sorafenib, a multi-kinase inhibitor, on apoptosis, NAD salvage, mitochondrial function and related signaling cascades in HCC cell lines. They demonstrated that sorafenib induced cellular apoptosis and decreased NAD concentrations in a dose dependent manner. Specifically, mitochondrial respiration and ATP levels were suppressed by sorafenib treatment in HUH7 cells, which was accompanied with an activation of AMPK. In addition, overexpression of SIRT1 could reverse the sorafenib-induced apoptosis by deacetylation of p53, indicating an underlying mechanism of resistance to sorafenib treatment in HCC. This manuscript is well written and the results are concisely presented. However, specific comments to further improve the manuscript are as follows.

1.      The rationale of p-ERK detection was not sufficient for the manuscript topic. As the title indicates the main findings in this manuscript, sorafenib could induce apoptosis in hepatocellular carcinoma cells. To better understand the sorafenib-induced apoptosis in HCC cells, an active form of caspase-3 could be used for the detection of apoptotic events. The detailed apoptotic pathway could be explained in the discussion, via extrinsic or intrinsic pathways.

2.      For the statistical analysis, the authors claimed the data for serum free medium or control (fold) was set 1, why do some figures have the error bars in controls, some figures do not have?

3.      Densitometric analysis of SIRT1 should be performed in Figure 6. HUH7 cells showed significantly reduced SIRT1 protein levels after incubation with sorafenib (Figure 2D). Why was this finding not reproduced in Figure 6A?

4.      In figure 3 legend, Sorafenib decreases activation of the AMPK pathway in HUH7 cells, which is inconsistent with the western blot data (Figure 3E).

5.      In line 66. (e.g.) The addition of the "or not" with “whether” is neither logically nor grammatically. The author should modify all in the manuscript.

6.      In line 46 "an" is "a";

7.      Cell lines information should be noted in the figures (e.g. Figure1 B, C, D).

8.      The molecular weight of protein should be noted in the figure 2,3,4,5,6 as figure 1.

Reviewer 2 Report

This Manuscript led by Antje Garten et al., entitled “Sorafenib-induced apoptosis in hepatocellular carcinoma is reversed by SIRT1” showed that sorafenib influences the NAD/SIRT1/AMPK axis in hepatocellular carcinoma. The paper is well written and interesting to read, however  the below point-by-point comments will help improve the manuscript.

-Please include the analysis of phosphorylation of ERK after stymulation with Sorafenib, and LDH release, also for HepG2  cells (not only cell viability and apoptosis).

-Please perform  the analysis of Figure 2, also for HepG2 cells.

-Please clearly specify the conclusions and place them in a separate chapter.